# Emotion Classification and Achievement of Students in Distance Learning Based on the Knowledge State Model

**Yahe Huang** [1,*,†] **and Dongying Bo** [2,†]

1   School of Traditional Chinese Medicine, Jinan University, Guangzhou 510632, China
2   School of Mechanics and Construction Engineering, Jinan University, Guangzhou 510632, China
*   Correspondence: hyh@jnu.edu.cn
†   These authors contributed equally to this work.

**Abstract:** Since the outbreak of COVID-19, remote teaching methods have been widely adopted by schools. However, distance education can frequently lead to low student emotional engagement, which can not only cause learning burnout, but also weaken students' interest in online learning. In view of the above problems, this study first proposed a learner knowledge state model that integrates learning emotions under the background of digital teaching to accurately describe the current learning state of students. Then, on the basis of the public face dataset Iapa, we built an online multi-dimensional emotion classification model for students based on ResNet 18 neural network. Experiments showed that the method has an average recognition accuracy of 88.76% for the four cognitive emotions of joy, concentration, confusion, and boredom, among which the accuracy of joy and boredom is the highest, reaching 96.3% and 97.0% respectively. Finally, we analyzed the correlation between students' emotional classification and grades in distance learning, and verified the effectiveness of the student's emotional classification model in distance learning applications. In the context of digital teaching, this study provides technical support for distance learning emotion classification and learning early warning, and is of great significance to help teachers understand students' emotional states in distance learning and promote students' deep participation in the distance learning process.

**Keywords:** distance learning; sentiment classification; knowledge state model; ResNet 18 neural network

## 1. Introduction

The COVID-19 outbreak that began in 2019 has greatly affected the economic life of society. According to current teaching arrangements, students preview classes by watching a course video, participate in the teacher's courses through remote check-in to form attendance records, complete exercises to consolidate knowledge, and r collaborate to build knowledge through interactive communication such as discussion, "likes", and evaluation. However, long-term online learning faces difficulties, which include low continuous participation and poor interaction among students. In the context of digital teaching, related research on educational data mining and intelligent teaching assistance systems has become a hot issue in the field of educational technology in recent years [1]. Accurate perception of students' emotional states in online learning is crucial to personalized learning and is a prerequisite for cracking the "emotional deficit" in current online education [2]. In recent years, the continuous improvement of Internet of Things technology, sensor technology, and big data computing power has provided technical support for learning behavior analysis. Learning emotion detection based on intelligent technology has also attracted widespread attention from scholars [3]. Expression is not only the external manifestation of the activities of the inner world but also a physiological response to the external world [4]. In the actual classroom-style teaching environment, teachers can observe the subtle changes

in students' faces to infer the students' inner world. For instance pouting their mouths may express disgust for the teacher's teaching content [5], closing their eyes may indicate that the learners are in a state of fatigue [6], and so on. In classroom teaching, teachers and students can keep abreast of students' learning status through face-to-face communication. However, in distance teaching, due to the separation of time and space, teachers know nothing about students' learning emotions and cannot provide students with targeted teaching strategies. Therefore, it is critical to add an emotional cognition function to the network teaching system [7].

In recent years, the effectiveness of computer vision-based emotion recognition methods has been extensively verified. Fatahi et al. developed a self-assessment system with emotional cognition, which can analyze learners' various emotional states and cognitive situations during the learning process [8]. Sloep et al. established a personalized network learning system by studying the learning behavior, learning emotion, and learning state of learners in the network teaching system [9]. Professor Noori F designed a network teaching model that considers learners' personality characteristics and emotions [10]. BITS at the University of Regina in Canada has developed an intelligent teaching system that provides intelligent tutoring for primary learners using Bayesian technology [11]. Boban et al. proposed a personalized teaching model based on learning style recognition and a hybrid recommendation teaching strategy, which can provide learners with personalized teaching strategies [12].

To sum up, there have been significant advances in the automatic perception of emotion in distance learning. The learning state of learners can be predicted based on the real situation of each facial feature. Finally, according to the predicted results, an adaptive learning system is adopted to automatically adjust learning strategies, recommend learning resources, and provide early warnings. However, there are still many problems in the research on emotion classification of distance-learning students based on knowledge state model. For example, emotion recognition models are less accurate in identifying emotions such as pleasure, concentration, confusion, and boredom. Facial video-based emotion recognition methods have a lot of room for improvement. The use of single mode instead of fusion multi-mode signals leads to weak generalization ability of the model, and the knowledge state model is not widely used. In this study, a learner knowledge state model integrating learning emotions under the background of digital teaching is proposed, and a multi-dimensional online classification model of students' emotions is constructed based on ResNet 18 neural network. Two modules of feature embedding and feature aggregation are added to identify students' cognitive emotional states in distance learning. Finally, this study analyzes the correlation between the emotion classification of distance learning students and their grades, and verifies the validity of the emotion classification model in distance learning.

## 2. Related Works

Psychologists' research shows that emotions greatly affect perceptual choices, memory, and thinking activities in the human cognitive process [13]. Cognitive activities, such as causal reasoning, goal evaluation, and planning processes, all accompany emotion [14]. Therefore, sensing the emotions of learners and providing timely positive help can effectively improve the cognitive ability of learners [15].

Pekrun first proposed the concept of learning emotion in 2002 and defined it as an emotion that is directly related to learning, teaching, academic achievement, etc. [16]. Pekrun also proposed the control value theory to analyze the relationship between achievement and learning emotion [17]. Efklides encodes seven learned emotions, including pleasure, curiosity, confusion, anxiety, depression, boredom, and surprise [18]. Critcher believes that learning is an "attention-emotion" model that combines cognitive and emotional experience, and that emotion has a direct impact on the human learning process and other functions [19]. Kort proposes a comprehensive four-quadrant model that explicitly links learning and affective states. This theory divides academic emotions into four types with

valence as the horizontal axis and learning degree as the vertical axis. A typical learning experience includes a series of emotional experiences, that is, with the deepening of learning, emotions should fluctuate [20]. Ryan studied the frequency, persistence, and impact of boredom, frustration, confusion, preoccupation, happiness, and surprise among students with different populations, different research methods, and different learning tasks as variables [21].

In recent years, educational researchers have measured the emotional state of students based on data mining, computer vision, and other methods, and the effectiveness of computer vision-based emotion recognition methods has been extensively verified. Myers built a convolutional neural network to detect the three emotional states of boredom, engagement, and neutrality of students [22]. Based on facial expressions and heart rate information in videos, Chen detected learners' emotions during the writing process and achieved significant improvements in robustness and accuracy [23]. It can be seen that learners experience many different types of emotions in the learning process. The differences in the valence and learning level of these emotions and the emotional state is not fixed and changes with the learning process. The influence of emotion on learning has also been widely noted, and Rowe et al. have done some empirical studies to confirm that learning emotion is related to differences in the performance of learners in the short term [24].

In addition to this, earlier studies attempted to explain human behavior during learning by understanding learners' emotional and cognitive processes, proposing the concept of a knowledge state model. For example, Alepis proposed a multi-criteria theory combined with bimodal emotion recognition and applied it to the mobile education system [25]. Yang proposed an accurate definition of the knowledge state model. He believed that the dimensions of the knowledge state model should correspond to various aspects of the students in the real learning environment, and the attributes of the knowledge state model should represent the characteristics of the students in the real learning environment [26]. Pavlik believed that the knowledge state model represented the characteristics and corresponding levels of students, including knowledge skills, cognitive behavior, emotional experience, etc. [27]. Bontcheva proposed that knowledge modeling is inseparable from knowledge features, such as the definition of knowledge components contained in knowledge points, the mapping between projects and these knowledge components, and project difficulty, etc. [28]. Abyaa believed that those constructing a knowledge state model should first identify and select the appropriate characteristics that affect learners' learning, then consider the learner's mental state in the learning process, and finally select the appropriate modeling technology to simulate the optimal state of each feature [29].

The above studies contain some innovations in content and perspective, but they represent preliminary exploratory research, in terms of a knowledge state model that integrates learning emotions, and there are still many deficiencies. Due to the complexity of learning emotions and the diversity of learner models, this study focuses on the following three issues:

(1) How to effectively classify students' emotions according to facial images in distance learning?
(2) How to build a knowledge state model integrating learning emotion?
(3) How to explore the characteristics of knowledge state model integrating learning emotion?

## 3. Student Emotion Classification

### 3.1. Student Emotion Classification Algorithm

This study focuses on the emotional classification goals of students in the distance learning process, analyzes the sequence information of students' facial videos in the online learning process, uses a frame attention network to extract the frame level features closely related to students' emotions, and judges students' emotional states in the classroom, such as pleasure, concentration, confusion, and boredom. We used the face public dataset lapa-

dataset (https://github.com/JDAI-CV/lapa-dataset, accessed on 2 November 2021) to train the model. Due to the rich emotional information expressed by faces [30], videos contain more dynamic emotional changes than static images and focus on image frame features containing more emotional information, which can improve the accuracy of emotion recognition. Therefore, this study uses video data and deep learning methods to mine facial features to detect students' emotions during distance learning. The overall flow chart of the emotion classification algorithm based on the student facial video sequence is shown in Figure 1.

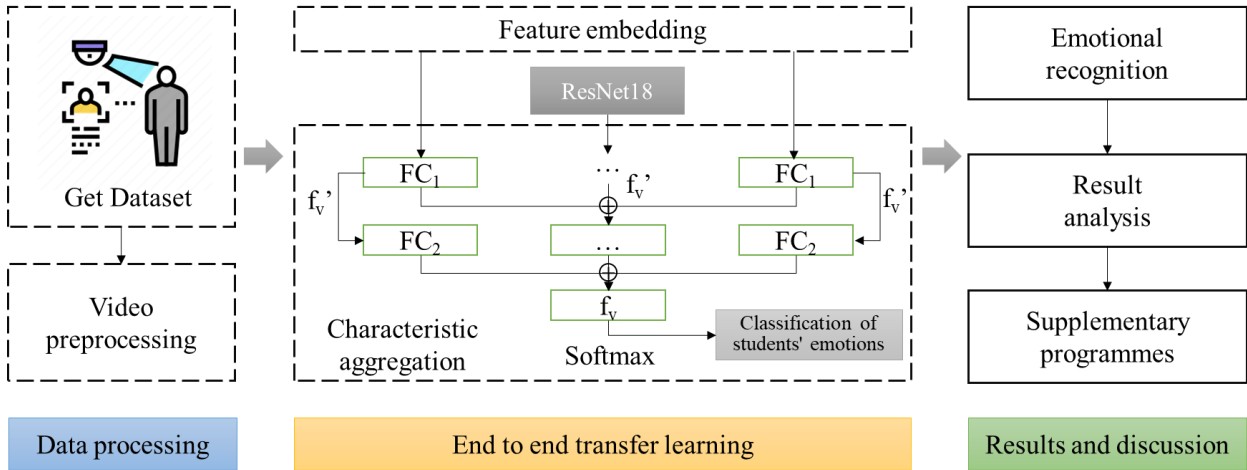

**Figure 1.** Overall flow chart of emotion classification algorithm based on student facial video sequence.

### 3.2. Module of Feature Embedding and Feature Aggregation

Although the video contains more dynamic features, not all frames are conducive to emotional discrimination [31]. Therefore, this study uses a frame attention network to recognize students' online learning emotions. The frame attention network structure is shown in Figure 2. The network includes two modules: feature embedding and feature aggregation. In the feature embedding module, feature representation is generated for all image frames sampled from the video, and these features are transferred to the feature aggregation module; In the feature aggregation module, there are two attention blocks. Through these two attention blocks, all the frame features of the video are aggregated to get a compact video feature representation, which can distinguish the four cognitive emotions of students: pleasure, focus, confusion, and boredom.

### 3.2.1. Feature Embedded Module

In the feature embedding module, ResNet 18 model is used as the basic network structure to complete the extraction of image features. In order to add the attention module, this study modifies the last output layer of ResNet 18, and replaces the last average pooling layer of ResNet 18 with a binary adaptive mean aggregation layer to perform global average pooling. In addition, by squeeze compression of spatial features, a 512 dimensional feature representation is embedded for each image frame. Specifically, for video $V$ of a $t$ frame, the video frame is represented as $\{I_1, I_2, \ldots, I_t\}$, and the features extracted from each frame using ResNet 18 network are represented, as all these frame features with 512-dimensional feature vectors are input to the feature aggregation module for feature aggregation.

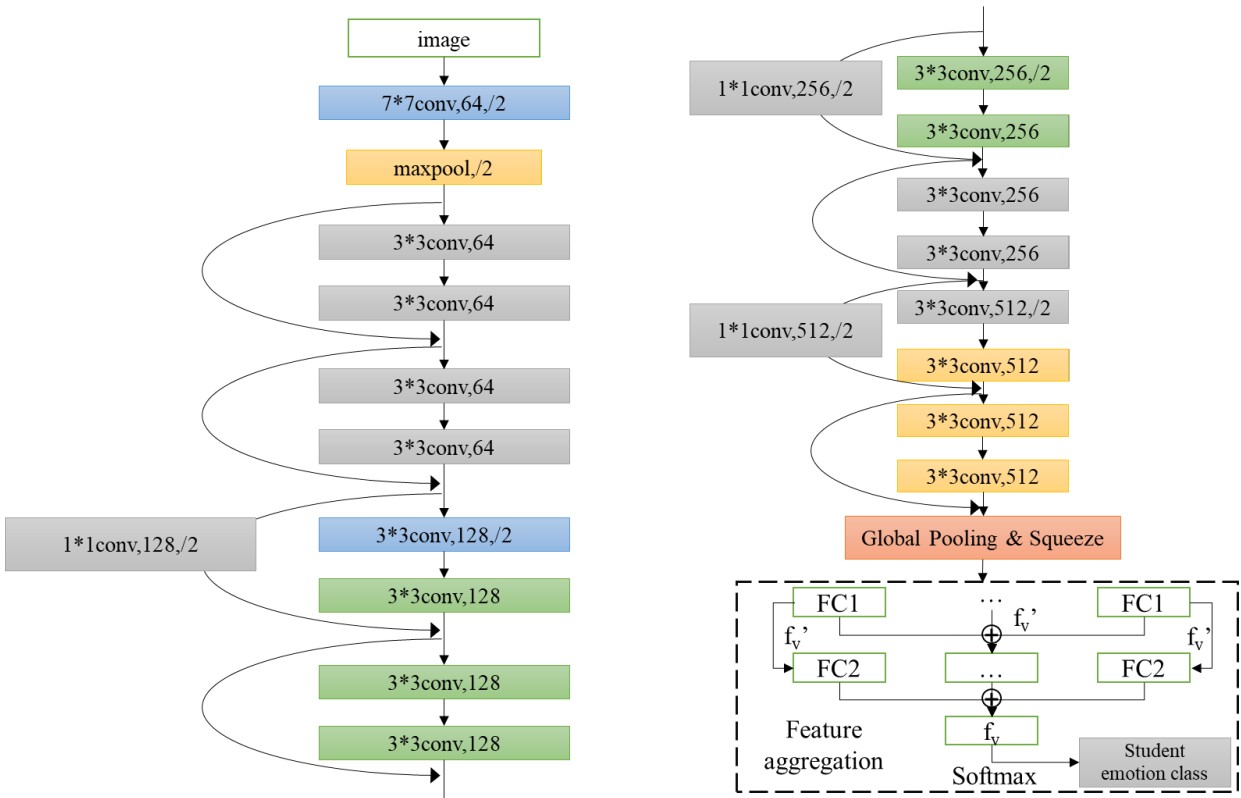

**Figure 2.** Frame attention network structure.

### 3.2.2. Feature Aggregation Module

Frame feature aggregation module is the key module of frame attention network. For a video $V$, all frame features from the feature embedding module learn two attention weights based on two attention blocks. These are the self-attention weight from a single frame image and the relationship between the image frame and the global attention.

For a single frame feature, a full connection layer and sigmoid activation function are used to initialize the self attention weight, which can be expressed by Formula (1):

$$\alpha_i = \sigma\left(f_i^T q^0\right) \tag{1}$$

Here, $\sigma$ is the sigmoid activation function, and is expressed in the following Formula (2):

$$\sigma(x) = \frac{1}{1 + e^x} \tag{2}$$

$q^0$ is a parameter of the full connection layer. Based on the frame features and the self attention weight of each frame, all the input frame features are aggregated to obtain the feature of the video sequence as Formula (3):

$$f_v' = \frac{\sum_{i=1}^n a_i f_i}{\sum_{i=1}^n a_i} \tag{3}$$

Here, $f_v'$ is used as a fusion feature to distinguish the cognitive emotion of online learning. The learning of self attention weight only uses a single frame feature, using a nonlinear mapping method, and the weight acquired through this method is still rough. Because $f_v'$ aggregates the features of the whole video, you can model the frame feature and the global feature $f_v'$ to further refine the weight of attention. By connecting the global features with the frame features, and by activating a full connection layer, a new relational attention weight is estimated for the image frames.



Based on this method, the attention weight of the i-th frame can be expressed as the following Formula (4):

$$\beta_i = \sigma\left(\left[f_i : f_v'\right]^T q^1\right) \tag{4}$$

Here, $q^1$ is the parameter of the full connection layer; $\sigma$ is a sigmoid activation function. Finally, according to the self attention weight and relational attention weight, all image frame features can be aggregated into a new video feature expression Formula (5):

$$f_v = \frac{\sum_{i=1}^n a_i \beta_i [f_i : f_v']}{\sum_{i=1}^n a_i \beta_i} \tag{5}$$

the video feature vector $f_v$, obtained by aggregation, is used to identify students' online learning emotions.

### 3.3. Transfer Learning Based on ResNet Pre Training

In this study, a ResNet 18 model pre-trained on MSCeleb-1M face recognition dataset [32] and FER+expression dataset [33] is used for transfer learning.

### 3.3.1. Video Preprocessing

This research uses ffmpeg library [34] to sample key frames from students' emotional videos, samples each emotional video at a frequency of 10 frames per second, and generates a variable number of frame sequences for each emotional video. Since the cognitive emotion database constructed in this study is collected in the student classroom, and the interference factors such as light and background are not controlled, the collected data contains various complex backgrounds. In order to successfully carry out the subsequent cognitive emotion detection task, it is necessary to preprocess the frame sequence. In order to accurately locate the face and eliminate redundant spatial information such as background, this research uses OpenCV and Dlib library.

First, we used OpenCV [35] to read image frames and dlib face detector [36] to locate faces. Then, we used 51 facial feature points to align the human face. When dlib was used for face alignment and clipping, 0.25 padding was set. The feature points after padding are more compact, so an aligned image containing more face areas can be obtained. The aligned face was cropped to a 224 × 224 picture. Using this method, the face images of all frame sequences were detected and clipped.

### 3.3.2. Data Loading

A special loading method was set for data loading. The specific operations are as follows:

For the video $V$ of a $t$ frame image, the frame sequence can be expressed as $\{f_1, f_2, \ldots, f_t\}$. All frames are divided into $x$ groups in chronological order. One frame image is sampled from $x$ groups each time to obtain a sample of $x$ frames. Samples are randomly sampled $t$ times to obtain $t$ samples of $x$ frames. Since each sample is sampled from different groups in chronological order, the samples contain the time information of the video. Moreover, in this way, the sample is expanded by $x$ times, which makes the training data increase significantly. Different values of $x$ can obtain a different number of samples, and $t$ is set to 3 by default. This study compares the effects of models trained based on different frequency $x$-sampling data sets.

### 3.3.3. Activation Function and Loss Function

The video-based student cognitive emotion recognition algorithm selects Relu as the activation function. First, Relu can achieve unilateral inhibition, so that the network can selectively move forward. When input $\leq 0$, the neuron is in a deactivated state, and when input $> 0$, the neuron is activated. Second, Relu has sparse activation, and its convergence speed is faster than other activation functions. In addition, the relatively broad excitation boundary enables Relu to make neurons with input $> 0$ active.

Deep learning consists of network structure, algorithm, strategy, and other elements. When the network structure and algorithm are selected, appropriate strategies need to be selected to train network parameters to obtain the optimal model; because students' emotion recognition is essentially a multi-classification problem, cross entropy loss function is used. This is because the greater the error of cross entropy loss function calculation, the greater the gradient, and the faster the convergence.

The calculation formula of cross entropy function is:

$$C = -\frac{1}{n} \sum x[y \ln a + (1-y)\ln(1-a)] \tag{6}$$

In the above formula, $x$ represents the sample, $y$ represents the real label, $a$ represents the model output, and $n$ represents the total number of samples.

The training of deep learning is actually the adjustment of network parameters. The gradient descent method is used to find the fastest direction of gradient descent and speed up the convergence of loss function. In this study, The method of random gradient descent was selected. It divides training samples into several batches, trains batch size data samples in one iteration, calculates loss, and updates weights through the training of one batch. After batch iterations, an epoch training was completed. In the stochastic gradient optimization algorithm, the choice of batch size has an impact on the performance of the model.

## 4. Knowledge State Model with Learning Emotion

This study establishes a learner knowledge state model that takes learners' learning emotions into consideration, which can effectively simulate real learning situations, be closer to learning reality, make more accurate predictions of learners' knowledge state, reflect students' current learning state, provide a more accurate basis for action for real educators to conduct academic early warnings and adjust teaching strategies, promote the realization of personalized teaching methods, and provide future education big data. The development of an intelligent teaching assistant system provides new modeling methods and practical reference.

Assume that the logical function of the correct probability of the learner's nth attempt is:

$$p_{ij} = \frac{1}{1 + e^{-s_{ij}}} \times \frac{1}{1 + e^{-z_{ij}}} \tag{7}$$

where,

$$z_{ij} = \theta_i + \sum_{k \in KCs} q_{jk}(\beta_k + \gamma_k \times opp(k,i)) \tag{8}$$

Emotions are expressed as:

$$S_{ij} = w_0 + \sum_{k \in KCs} w_k x_k \tag{9}$$

In this model, students' emotion classification modeling in distance learning is used as a weight variable for learning results. $X_k$ represents one of the emotions of pleasure, focus, confusion, and boredom, and $w_k$ represents the weight of the corresponding emotion. Emotion $X$ uses Boolean representation to indicate the possible influence of emotion on learning results in the learning process. For example, the learners' confusion emotion $X$ (1 = confusion; 0 = no confusion) in the learning process is expressed as $S_{ij} = w_0 + w_1 x_1 + \cdots + w_k x_k$ in the learner model. W means that the possible influence of learners' confusion on knowledge point learning will change with the changes in learners' individual differences, number of learning items, and learning state.

Worcester Polytechnic Institute. This data set contains data on learning behavior and learning emotion. This data set was used to test the effectiveness of the proposed model.

Students' behavior data and emotional data in the learning process were refined, and five learning characteristics were selected.

The names and corresponding descriptions of the learning characteristics are as follows:

Analytical view: this feature indicates that learners actively click during the learning process;

Correct or incorrect first attempt: this feature indicates whether the learner's first attempt at solving problems is correct or not;

Number of learning knowledge points: this feature represents the number of different knowledge points learned by learners;

Total attempts: this feature represents the total attempts of learners in the learning process;

Confusion emotion: this indicates the average probability of each confusion emotion.

After defining the above learning characteristics, this study compiled statistics on the frequency of the above five characteristics and learners' learning time respectively, and finally obtained a data set containing information such as student number, five learning characteristics, and learning duration.

When the the samples with missing learning data of a single subject at the same knowledge point are removed, we obtain the data description of students' learning behavior, as shown in Table 1.

**Table 1.** The description of behavior data.

|  | Total | Average Value | Maximum | Minimum |
| --- | --- | --- | --- | --- |
| Times of confusion | 197 | 7.04 | 19 | 2 |
| View resolution times | 322 | 2.875 | 6 | 0 |
| Duration | 27,311 | 975.39 | 1543 | 642 |
| Correct times | 253 | 9.04 | 14 | 5 |

Through the analysis of 543 pieces of data, we can see that students felt confusion 197 times in the process of answering questions. The average number of perplexities of a single student was 7.04, and the average number of correct answers was 9.04. These two values are relatively low. This may be because the students do not fully understand the knowledge points investigated, do not feel confused, or do not choose the correct answer in the process of answering. However, this does not rule out the possibility that that the students are in a negative state in the process of learning.

This experiment set a specific data sample loading method. Each sample contained T frames. For video sampling, T was set to 3 by default during training. In this study, cross entropy loss was selected as the loss function of model training, and random gradient descent algorithm was used to optimize. Momentum was set to 0.9 and weight attenuation was $1 \times 10^{-4}$. The principle of random gradient descent algorithm can be expressed as follows:

$$w_{t+1} = w_t - \eta \frac{1}{n} \sum \nabla l(x, w_t) \tag{10}$$

$n$ is the batch size, $\eta$ is the learning rate. It can be seen that learning rate and batch size are two very important parameters that affect gradient descent. In order to get the best model, this study conducted parameter adjustment experiments, and trained 100 epochs for each parameter setting. In order to save the best model in the training, each epoch was trained, and the verification set was used to verify once. When the accuracy of the verification set increased, the model was saved, and the best model obtained from the training was selected for evaluation on the test set. For learning rate, $1 \times 10^{-4}$, $4 \times 10^{-4}$, and $1 \times 10^{-5}$ were set successively. For batch size, 8, 32, and 64 were set. The result of parameter adjustment is shown in Figure 3.

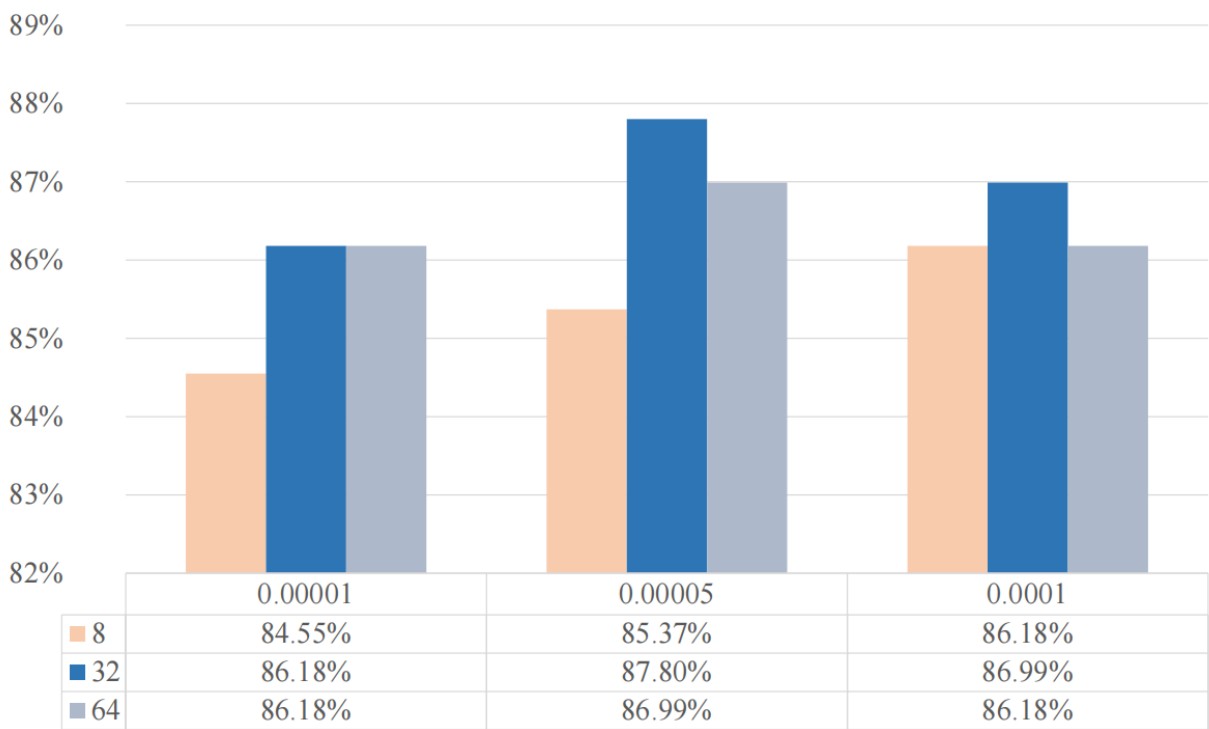

| | 0.00001 | 0.00005 | 0.0001 |
|---|---|---|---|
| 8 | 84.55% | 85.37% | 86.18% |
| 32 | 86.18% | 87.80% | 86.99% |
| 64 | 86.18% | 86.99% | 86.18% |

**Figure 3.** Experimental results of adjusting super parameters.

The abscissa represents different learning rates, and the ordinate represents different batch sample sizes. It can be seen that when the learning rate is set to $1 \times 10^{-5}$ and the batch size is 32, the training model has the highest accuracy. After parameter adjustment, the best model is saved.

On the basis of the classical knowledge state model, this study proposes a knowledge state model that integrates learning emotions, adjusts the parameter composition in the algorithm, and uses the linear combination of parameters to represent the characteristics of learning emotions, making up for the lack of consideration of the basic knowledge state model in the learners' learning emotions.

## 5. Analysis Results and Discussion

### 5.1. Experimental Results Based on Different Frame Feature Aggregation Strategies

This study classified dynamic features based on video level data, analyzed four cognitive emotional states of students in online learning environment, and compared three methods of video frame aggregation: one without attention weight, one with frame feature aggregation using self attention weight alone, and one with aggregation combining self attention weight and relational attention weight in time *T*-dimension dynamic feature aggregation. On the basis of fixed parameters and other experimental conditions, we only changed the frame feature aggregation, and compared the performance of the three fusion methods to the classification model. This experiment was set as follows: the data set division ratio was 8:1:1, the batch size was 32, the learning rate was set to $1 \times 10^{-4}$. The random gradient descent method was used for training, and the training iteration epoch was 100. Figure 4 shows the model evaluation results based on different frame aggregation strategy training.

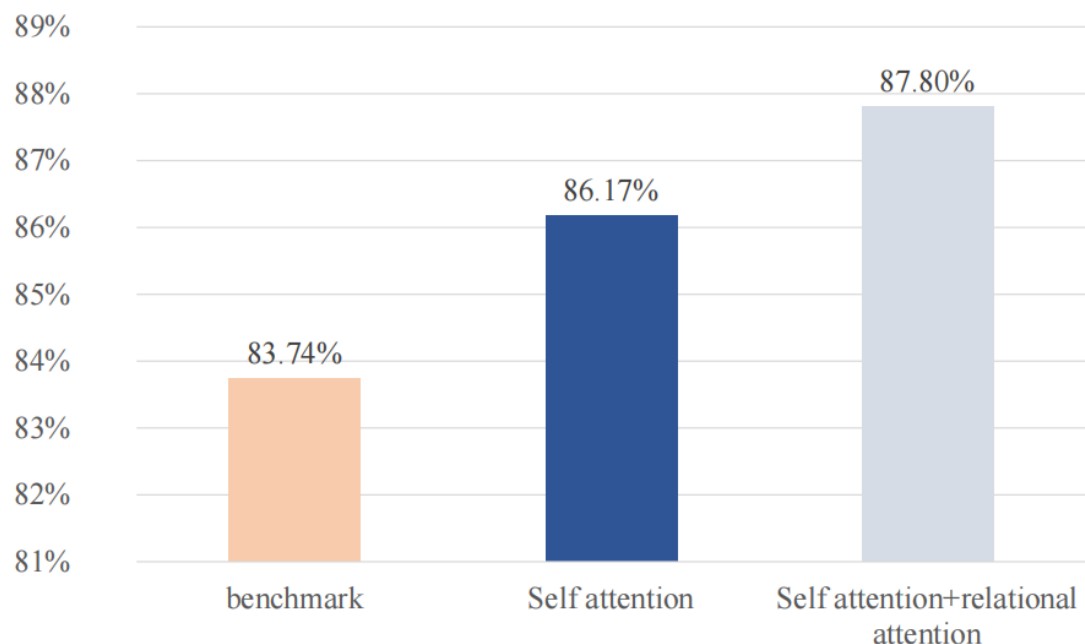

**Figure 4.** Results of different frame aggregation strategies.

It can be seen that combining the self attention weight and relational attention weight of image frame features is the best method among the three, and the accuracy of the model verification reaches 87.8%. Compared with the unweighted fusion, it is about four percentage points higher. This shows that it is effective to use frame attention network to recognize students' cognitive emotional state, which improves the accuracy of students' emotional recognition.

*5.2. Experimental Results of Different Sampling Frequencies*

In this study, we designed a special way to load data. For frame sampling in video, the video was first divided into $T$ groups, and then a frame from each segment was randomly selected. The sampled video frames were spliced in chronological order. Based on different sampling frequencies $T$, the time data length of the samples obtained was $T$, and for a video containing $X$ frames, the sample size obtained from the video was $T * X$. Therefore, the sampling frequency of data affected the total number of data samples. In addition to the default setting of $T = 3$, this study set $T$ to 2, 6, and 9 respectively, for comparison. Due to the experimental configuration, when $T = 9$, the experiment failed, due to insufficient memory of the device. Therefore, this experiment only compared $T = \{2, 6\}$. The experimental results show that the sampling frequency $T$ has no effect on the training of the model. Figure 5 shows the results of this comparative experiment.

Through experiments, the best performance cognitive emotion recognition model was obtained, and its recognition accuracy for four cognitive emotion states reached 87.8%. The following Figure 6 shows the recognition accuracy results of the best model for different emotion categories for further analysis.

This study compares the performance of the improved model and the traditional model through the learning curve. The results are shown in Figure 7. The results show that when students are learning effectively, their confusion is positively related to their learning results, while when they are not learning or learning ineffectively, their confusion is negatively related to their learning results.

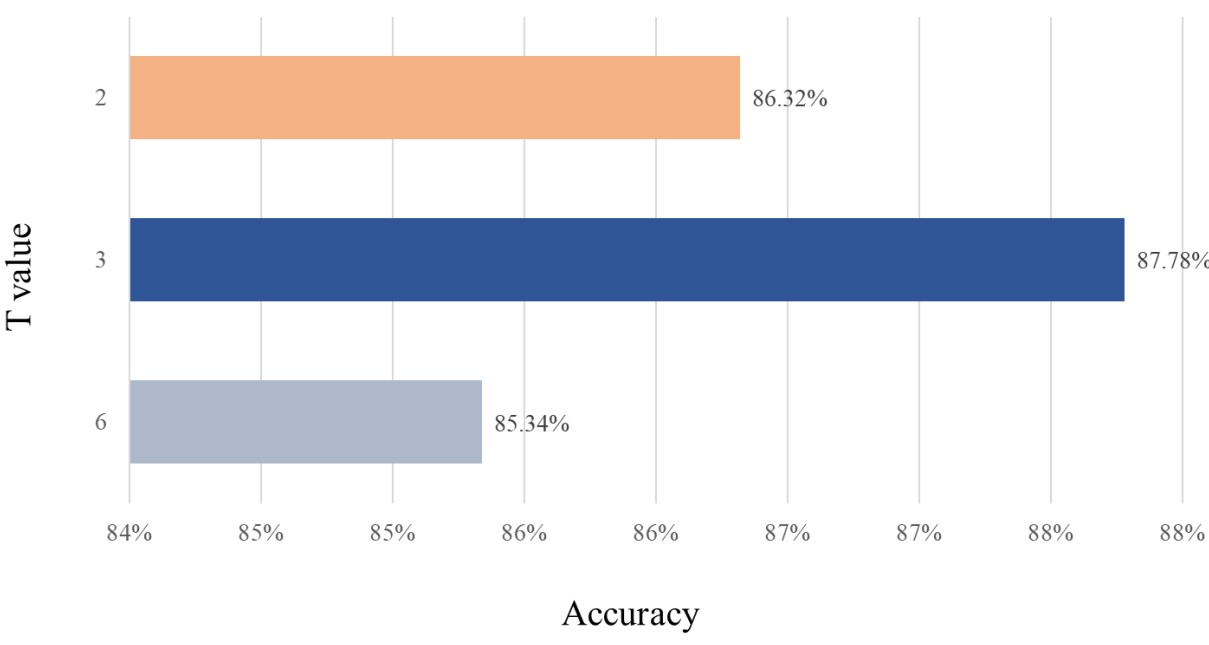

**Figure 5.** Model Accuracy at Different Sampling Frequencies *T*.

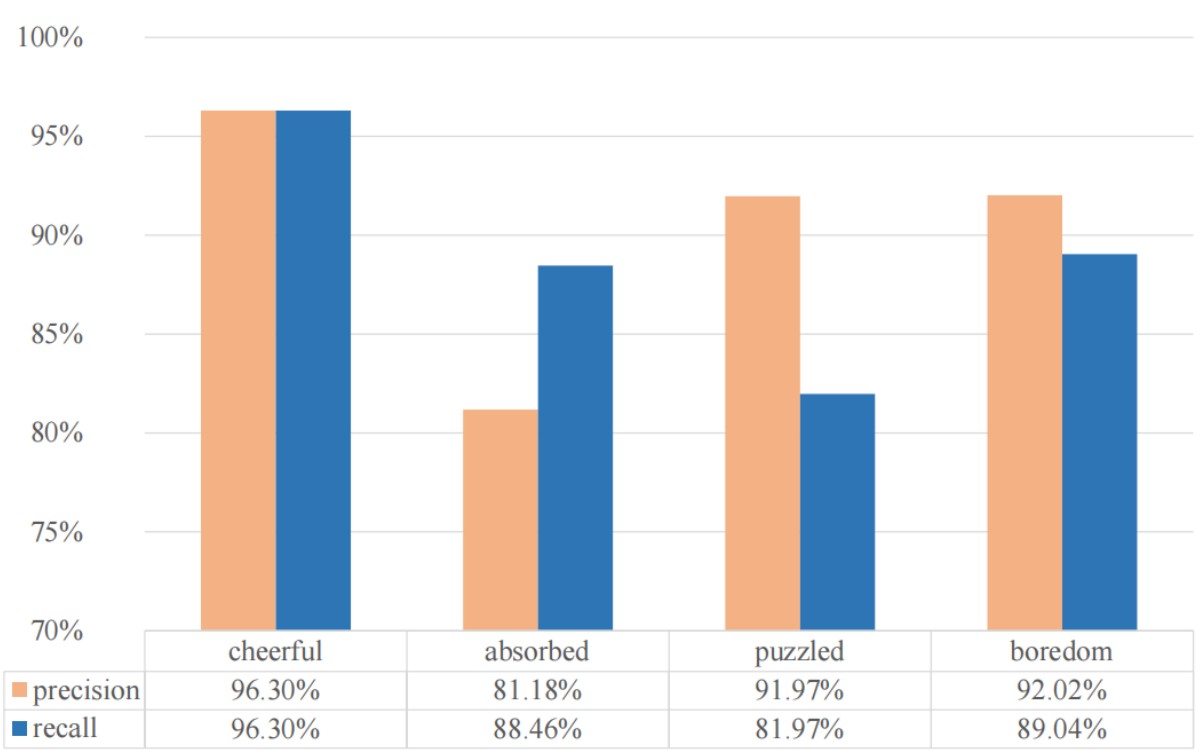

**Figure 6.** Recognition accuracy of different emotional states.

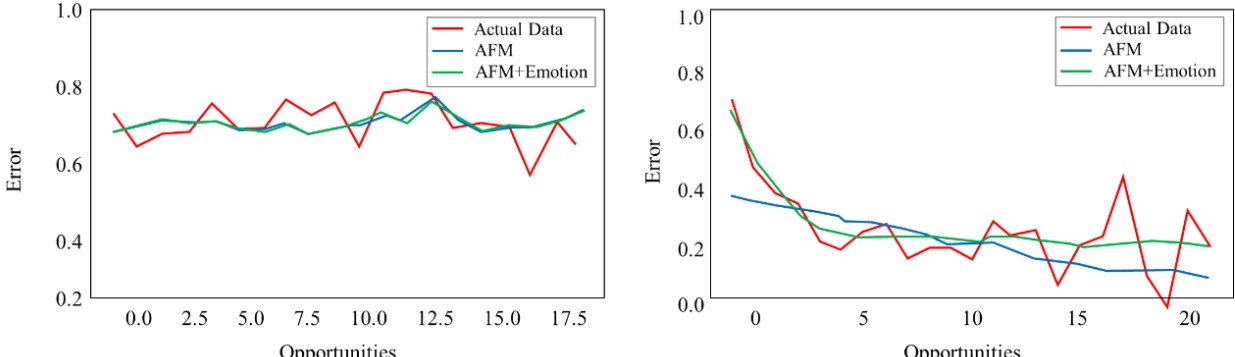

**Figure 7.** Comparison of learning curves of two knowledge points selected.

*5.3. Discussion*

Through the above research results, we can find that

(1) The learning curve drawn by the knowledge state model with learning emotion is closer to the real data curve, and the corresponding learning curve is more gentle.

(2) Happiness and boredom had the highest accuracy, reaching 96.3% and 97.0%, respectively. Corresponding to the commonly used two-dimensional emotional model, these two kinds of emotions are located at two arousal levels respectively, and are also the emotional states easily recognized by human visual perception. During the COVID-19 pandemic, these two types of emotions have been prominent among students in distance learning. However, the recognition accuracy of focus and confusion is low. This is mainly because facial expressions in the two states are similar and the data are not very different. Therefore, it is important to determine how to design the corresponding algorithm to improve the two kinds of online learning emotional states.

(3) Self-attention weight combined with relevant attention weight of image frame features is the best method, and the accuracy of model verification reaches 87.8%. It is about four percentage points higher than unweighted convergence. This indicates that it is effective to use frame attention network to identify students' cognitive emotional states and improve the accuracy of students' emotional recognition.

(4) The rate of students' learning errors decreases, showing a continuous learning state. At this time, the learners' knowledge state model integrating confusion emotions performs well, which verifies that if learners can effectively regulate their own confusion emotions and solve them when confusion emotions occur, confusion will successfully transition to a positive state. During the COVID-19 pandemic, learners have emotional problems at different levels due to factors such as home isolation, thus influencing the effect of online learning. Most parents of students cannot return to work at present, and some have even lost their jobs. The anxiety caused by the great pressure on parents may lead to negative emotions and even psychological crisis for learners. Educators should pay attention to the subtle changes in learners' emotions during this special period, so as to effectively reduce the risk of complicated and arduous school openings after the COVID-19 pandemic has eased.

## 6. Conclusions

Under the background of digital teaching, the rapid development of data mining technology has laid a good technical foundation for personalized teaching. This study constructs an online classification model of students' multi-dimensional emotions based on ResNet 18 neural network, innovatively adds two modules of feature embedding and feature aggregation, and uses frame attention network to extract frame level features closely related to students' emotions. The average recognition accuracy of four cognitive emotions, namely pleasure, focus, confusion and weariness, reaches 88.76%. This study uses a mathematical modeling method to add emotional factors as parameters to the model-

ing of learners' knowledge state, analyzes the correlation between students' emotional classification and their scores in distance learning, verifies the effectiveness of students' emotional classification model in the application of distance learning, and finds that there is a significant correlation between focus, confusion, boredom and learning results. In addition, this paper puts forward a new direction of thinking about teaching intervention. The study provides technical support for distance learning emotion classification and early warning, which is of great significance in helping teachers understand students' emotional state in distance learning promptly, and in promoting students' in-depth participation in the distance learning process.

In order to improve the research findings, future work will include:

(1) Further analysis and discussion of the integration of other learning emotions and learner models, and finding more good emotional features that can predict learning results in the learning process;

(2) Learning more methods to improve models, realizing the potential of models that comprehensively consider multiple learning emotions, enhancing the completeness of models that are more suitable for online learning in secondary schools, improving the reliability of learner models that integrate learning emotions, and developing learner knowledge.

**Author Contributions:** Conceptualization, D.B.; Methodology, Y.H.; Formal analysis, Y.H.; Investigation, D.B.; Writing—original draft, Y.H.; Project administration, D.B. All authors have read and agreed to the published version of the manuscript.

**Funding:** This research received no external funding.

**Institutional Review Board Statement:** The study did not require ethical approval.

**Informed Consent Statement:** Informed consent was obtained from all subjects involved in the study.

**Data Availability Statement:** The data that support the findings of this study are available from the corresponding author upon reasonable request.

**Acknowledgments:** The authors thank the anonymous reviewers who have provided valuable advice for this article.

**Conflicts of Interest:** The authors declare no conflict of interest.

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
