# Peer review of "Emotion Classification and Achievement of Students in Distance Learning Based on the Knowledge State Model"

_sustainability, doi:10.3390/su15032367_

Round 1

Reviewer 1 Report

The author presents the article entitled “Emotion classification and achievement of students in distance learning based on the knowledge state model”

This paper proposes a learner knowledge state model that integrates learning emotions under the background of digital teaching to describe students' current learning state accurately and constructs an online classification model of multidimensional emotions of the students based on the ResNet neural network to identify cognitive and emotional states in distance learning.

The article presents the following concerns:

  • Introduction section: Please provide indicators about the learning methods in distance and e-learning techniques according to the state of the art. Also, the controversy is not clear.

  • Please consider articles from Sustainability journal, such as https://doi.org/10.3390/su122410367 or https://doi.org/10.3390/su142013230.

  • The objective of the manuscript is not clear. Also, there are several reports in the emotion classification for students in e-learning methods, such as –Imani, M., & Montazer, G. A. (2019). A survey of emotion recognition methods with emphasis on E-Learning environments. Journal of Network and Computer Applications, 147, 102423. -- What is the novelty of the proposed work? What is the novelty?

  • Results: Please provide a picture or evidence about the performance of the algorithm in a real scenario.

  • Please present the Results and Discussion in separated sections. 

  • In Discussion section, Authors should discuss the results and how they can be interpreted from the perspective of previous work. Also, please add a table that compares the findings of the work vs the reported in the state of the art

  • Add hyperlinks to tables, figures, and references.

  • It is necessary to describe the structure of the text at the end of the introduction 

  • Apostrophes must be avoided, for example: Psychologists', learners’, etc.

  • The text must be written in the 3rd person or passive voice. It is necessary to be consistent in all text. 

  • Figures must be vectorized to see the details 

  • The year of reference 5 is missing. 

  • It is necessary to review the Editorial Style Manual for Authors

  • Please add  symbols of registered trademarks.

  • Line 51 could be justified with: The impacts of covid-19 on technological and polytechnic university students in mexico; The impacts of covid-19 on technological and polytechnic university teachers

  • COVID-19 issues are not covered along the manuscript, although this is seen in the abstract. Then, consider these scenaries regarding educational vs covid-19 issues as the references: Teachers’ perception in selecting virtual learning platforms: a case of mexican higher education during the covid-19 crisis; Analysis of emergency remote education in covid-19 crisis focused on the perception of the teachers.

  • The phrase "in this study" constantly appears. Consider looking for synonyms.

The following misspelling should be checked:

  1. line 163: It appears that “in the video” may be unnecessary in this sentence. Consider removing it. 

  2. line 211: It appears that “so” may be may be unnecessary in this sentence. Consider removing it. 

  3. line 375: The phrase “On the basis of” may be wordy. Consider changing by “Based on”

  4. line 458:  The phrase “in a timely manner” may be wordy. Consider changing by “promptly” or “on time”

  5. line 466: It appears that you have improperly spaced some punctuation. 

Author Response

Reviewer 1

1.Introduction section: Please provide indicators about the learning methods in distance and e-learning techniques according to the state of the art. Also, the controversy is not clear.

Response:we added the indicators as follows:

Their names and corresponding descriptions are as follows:

Analytical view: this feature indicates that learners actively click during the learning process;

Correct or incorrect first attempt: This feature indicates whether the learner's first attempt at learning problems is correct or not;

Number of learning knowledge points: this feature represents the number of different knowledge points learned by learners;

Total attempts: This feature represents the total attempts of learners in the learning process;

Confusion emotion: It indicates the average probability of each confusion emotion.

2.Please consider articles from Sustainability journal, such as https://doi.org/10.3390/su122410367 or https://doi.org/10.3390/su142013230.

Response:We added this literature to the citation

3.The objective of the manuscript is not clear. Also, there are several reports in the emotion classification for students in e-learning methods, such as –Imani, M., & Montazer, G. A. (2019). A survey of emotion recognition methods with emphasis on E-Learning environments. Journal of Network and Computer Applications, 147, 102423. -- What is the novelty of the proposed work? What is the novelty?

Response:The innovation of this paper lies in the use of frame attention network to extract frame-level features closely related to students' emotions. Two modules of feature embedding and feature aggregation are added to distinguish the cognitive state and emotional state of distance learning students.

4.Results: Please provide a picture or evidence about the performance of the algorithm in a real scenario.

Response:We believe that such evidence is meaningless, and in fact, the results and discussions present our findings in detail.

5.Please present the Results and Discussion in separated sections.

Response:5.3 Discussion

Through the above research results, we can find that

(1) The learning curve drawn by the knowledge state model with learning  emotion is closer to the real data curve, and the corresponding learning curve is more gentle.

(2) Happiness and boredom had the highest accuracy, reaching 96.3% and 97.0%, respectively. Corresponding to the commonly used two-dimensional emotional model, these two kinds of emotions are located at two arousal levels respectively, and are also the emotional states easily recognized by human visual perception. But the recognition accuracy of focus and confusion is low. This is mainly because facial expressions in the two states are similar and the data is not very different. Therefore, how to design the corresponding algorithm to improve the two kinds of online learning emotional states is of great significance.

(3) Self-attention weight combined with relevant attention weight of image frame features is the best method, and the accuracy of model verification reaches 87.8%. It is about 4 percentage points higher than unweighted convergence. This indicates that it is effective to use frame attention network to identify students' cognitive emotional states and improve the accuracy of students' emotional recognition.

(4) The rate of students' learning errors has been decreasing, showing a  continuous learning state. At this time, the learners' knowledge state model integrating confusion emotions performs well, which verifies that if learners can effectively regulate their own confusion emotions and solve them in time when confusion emotions occur, confusion will successfully transition to a positive state.

6.In Discussion section, Authors should discuss the results and how they can be interpreted from the perspective of previous work. Also, please add a table that compares the findings of the work vs the reported in the state of the art.

Response:We added a review to the literature review and added Section 5.3 to analyze the experimental results.

7.Add hyperlinks to tables, figures, and references.

It is necessary to describe the structure of the text at the end of the introduction

Apostrophes must be avoided, for example: Psychologists', learners’, etc.

The text must be written in the 3rd person or passive voice. It is necessary to be consistent in all text.

Figures must be vectorized to see the details

The year of reference 5 is missing.

It is necessary to review the Editorial Style Manual for Authors

Please add  symbols of registered trademarks.

Response:The above problems have been modified in the manuscript. As for the format, I hope the editor can further confirm.

8.Line 51 could be justified with: The impacts of covid-19 on technological and polytechnic university students in mexico; The impacts of covid-19 on technological and polytechnic university teachers;

Response:Thank you for your comment, we have revised this.

9.COVID-19 issues are not covered along the manuscript, although this is seen in the abstract. Then, consider these scenaries regarding educational vs covid-19 issues as the references: Teachers’ perception in selecting virtual learning platforms: a case of mexican higher education during the covid-19 crisis; Analysis of emergency remote education in covid-19 crisis focused on the perception of the teachers.

Response:We've added something new that The COVID-19 outbreak that began in 2019 has greatly affected the economic life of the entire society. According to the teaching arrangement, students preview by watching the course video, participate in the teacher's courses through remote check-in to form attendance records, complete exercises to consolidate knowledge, and realize collaborative innovation of knowledge through interactive communication such as discussion, "like" and evaluation. However, long-term online learning has the difficulty of low continuous participation and poor interaction among students.

10.The phrase "in this study" constantly appears. Consider looking for synonyms.

The following misspelling should be checked

line 163: It appears that “in the video” may be unnecessary in this sentence. Consider removing it.

line 211: It appears that “so” may be may be unnecessary in this sentence. Consider removing it.

line 375: The phrase “On the basis of” may be wordy. Consider changing by “Based on”

line 458:  The phrase “in a timely manner” may be wordy. Consider changing by “promptly” or “on time”

line 466: It appears that you have improperly spaced some punctuation.

Response:The above corrections have been revised in the manuscript.

Reviewer 2 Report

There are no research hypotheses. They should be added. It is worth considering separating a chapter, e.g. research methodology. Hypotheses may be included in it (or in a literature review in relation to current scientific achievements).
Figure 3-4, 6, do not display decimal places (zeros) on the vertical (%) axis.

A detailed description of the data contained in Table 1 is redundant. Certain information can be underlined/highlighted. Conclusions resulting from the presented descriptive statistics should be recorded. There is no need to describe the table data in words. This information does not require a detailed description.

In abstracts and conclusions, the authors write about correlation. There is nothing about correlation in the presentation of the results. Figure 7 is insufficient to assess the existence of a statistically significant relationship. Confirmation of the existence of correlations requires appropriate tests.

These graphs should be made smaller (aesthetic matter).

There is no discussion of the results. This chapter should be added. Reference should be made to research hypotheses. Recent studies should be added to the analysis. Overall, the review of the literature is poor.

Lack of discussion of the results results in a low evaluation of the applications.

The authors indicate the source of the research problem, e.g. in the COVID-19 pandemic. There is no reference to this statement in the discussion.

The language side needs to be refined.

Author Response

Reviewer 2

1.There are no research hypotheses. They should be added. It is worth considering separating a chapter, e.g. research methodology. Hypotheses may be included in it (or in a literature review in relation to current scientific achievements).

Response:In this paper, video data of students in different task situations are collected, and an online multidimensional emotion classification model for students is constructed based on ResNet 18 neural network. Frame attention network is used to extract frame-level features that are closely related to students' emotions. research hypotheses does not seem to be applied in this article.

2.Figure 3-4, 6, do not display decimal places (zeros) on the vertical (%) axis.

Response:In fact, we ensure that all data is kept to two decimal places. Please check with your editor further to see if you need to make any changes.

3.A detailed description of the data contained in Table 1 is redundant. Certain information can be underlined/highlighted. Conclusions resulting from the presented descriptive statistics should be recorded. There is no need to describe the table data in words. This information does not require a detailed description.

Response:We have modified the data and analysis in Table 1.

4.In abstracts and conclusions, the authors write about correlation. There is nothing about correlation in the presentation of the results. Figure 7 is insufficient to assess the existence of a statistically significant relationship. Confirmation of the existence of correlations requires appropriate tests.

Response:This study compares the performance of the improved model and the traditional model through the learning curve. The goal of the learning curve is to simulate and understand the student's mastery of knowledge during the learning process. Instead of analyzing the results of statistical lines.

5.These graphs should be made smaller (aesthetic matter).

Response:In order not to affect the typesetting work of the journal, we did not adjust the size of the pictures.

6.There is no discussion of the results. This chapter should be added. Reference should be made to research hypotheses. Recent studies should be added to the analysis. Overall, the review of the literature is poor.

Response:We added a review to the literature review and added Section 5.3 to analyze the experimental results.

7.Lack of discussion of the results results in a low evaluation of the applications.

Response:We added Section 5.3 to analyze the experimental results

8.Through the above research results, we can find that

(1) The learning curve drawn by the knowledge state model with learning  emotion is closer to the real data curve, and the corresponding learning curve is more gentle.

(2) Happiness and boredom had the highest accuracy, reaching 96.3% and 97.0%, respectively. Corresponding to the commonly used two-dimensional emotional model, these two kinds of emotions are located at two arousal levels respectively, and are also the emotional states easily recognized by human visual perception. But the recognition accuracy of focus and confusion is low. This is mainly because facial expressions in the two states are similar and the data is not very different. Therefore, how to design the corresponding algorithm to improve the two kinds of online learning emotional states is of great significance.

(3) Self-attention weight combined with relevant attention weight of image frame features is the best method, and the accuracy of model verification reaches 87.8%. It is about 4 percentage points higher than unweighted convergence. This indicates that it is effective to use frame attention network to identify students' cognitive emotional states and improve the accuracy of students' emotional recognition.

(4) The rate of students' learning errors has been decreasing, showing a  continuous learning state. At this time, the learners' knowledge state model integrating confusion emotions performs well, which verifies that if learners can effectively regulate their own confusion emotions and solve them in time when confusion emotions occur, confusion will successfully transition to a positive state.

The authors indicate the source of the research problem, e.g. in the COVID-19 pandemic. There is no reference to this statement in the discussion.

Response:We've added something new that The COVID-19 outbreak that began in 2019 has greatly affected the economic life of the entire society. According to the teaching arrangement, students preview by watching the course video, participate in the teacher's courses through remote check-in to form attendance records, complete exercises to consolidate knowledge, and realize collaborative innovation of knowledge through interactive communication such as discussion, "like" and evaluation. However, long-term online learning has the difficulty of low continuous participation and poor interaction among students.

9.The language side needs to be refined.

Response:We have checked the sentences in the full paper.

Reviewer 3 Report

Thank you for submitting your research paper for sustainability. This is a well-written paper, but there are some issues the authors need to address.

1. The novelty of this paper is not clear. The authors need to clearly highlight the main contributions (both theoretical and practical) of their work. This can be achieved by identifying the research problem(s) and the gaps in the literature this research aims to fulfil.   

2. Both theoretical and practical implications are not highlighted.

3. There are some minor corrections in the structure of some paragraphs.

4. It is recommended to separate the discussion from the findings.

Author Response

Reviewer 3

1.The novelty of this paper is not clear. The authors need to clearly highlight the main contributions (both theoretical and practical) of their work. This can be achieved by identifying the research problem(s) and the gaps in the literature this research aims to fulfil.   

Response:At the end of the literature review, we add a new review to highlight the major contributions of this paper.

2.Both theoretical and practical implications are not highlighted.

Response:This study analyzes the correlation between emotion classification and achievement of students in distance learning and verifies the validity of emotion classification model in distance learning application. Under the background of digital teaching, this study provides technical support for emotion classification and learning early warning in distance learning. It is of great significance to help teachers to understand students' emotional state in time and to promote students' deep participation in distance learning process

This study focuses on the following three issues:

(1) How to effectively classify students' emotions according to facial images in distance learning?

(2) How to build a knowledge state model integrating learning emotion?

(3) Explore the characteristics of knowledge state model integrating learning emotion.

3.There are some minor corrections in the structure of some paragraphs.

Response:We have checked the sentences in the full paper.

4.It is recommended to separate the discussion from the findings.

Response:We added Section 5.3 to analyze the experimental results.

Through the above research results, we can find that

(1) The learning curve drawn by the knowledge state model with learning  emotion is closer to the real data curve, and the corresponding learning curve is more gentle.

(2) Happiness and boredom had the highest accuracy, reaching 96.3% and 97.0%, respectively. Corresponding to the commonly used two-dimensional emotional model, these two kinds of emotions are located at two arousal levels respectively, and are also the emotional states easily recognized by human visual perception. But the recognition accuracy of focus and confusion is low. This is mainly because facial expressions in the two states are similar and the data is not very different. Therefore, how to design the corresponding algorithm to improve the two kinds of online learning emotional states is of great significance.

(3) Self-attention weight combined with relevant attention weight of image frame features is the best method, and the accuracy of model verification reaches 87.8%. It is about 4 percentage points higher than unweighted convergence. This indicates that it is effective to use frame attention network to identify students' cognitive emotional states and improve the accuracy of students' emotional recognition.

(4) The rate of students' learning errors has been decreasing, showing a  continuous learning state. At this time, the learners' knowledge state model integrating confusion emotions performs well, which verifies that if learners can effectively regulate their own confusion emotions and solve them in time when confusion emotions occur, confusion will successfully transition to a positive state.

Round 2

Reviewer 1 Report

The manuscript can be accepted 

Author Response

Thank you for your comments and recognition of this manuscript.

Reviewer 2 Report

Some comments from the review have been included in the new version of the article. However, some comments were omitted or ignored.

Research questions do not replace research hypotheses. Research hypotheses have not yet been formulated. They should be formulated. Then you must clearly state whether they have been confirmed/rejected.

The discussion of the results still lacks a reference to the COVID-19 pandemic.

The aesthetics of the charts has not been improved. Still in charts 3, 4, 6 there are entries, e.g. 80.00% instead of 80% (as in chart 5).

The above comments do not interfere with the substantive content of the article. The scientific importance of the journal requires the authors to take into account the above comments.

The reviewer accepts the authors' explanations. The above comments are intended to improve the quality of the article. They are not obligatory.

Author Response

1.Research questions do not replace research hypotheses. Research hypotheses have not yet been formulated. They should be formulated. Then you must clearly state whether they have been confirmed/rejected.

Response: After consideration, the authors believe that the research hypothesis does not seem to need to appear in this article. In fact, this study built a multi-dimensional online classification model of students' emotions based on ResNet 18 neural network, innovatively added two modules of feature embedding and feature aggregation, and used frame attention network to extract frame-level features closely related to students' emotions. If the reviewer insists on this point of view, could you please provide some referable cases or more detailed tips.

2.The discussion of the results still lacks a reference to the COVID-19 pandemic.

Response: In the Discussion, we added new analysis based on the background of COVID-19 and highlighted them in blue.

3.The aesthetics of the charts has not been improved. Still in charts 3, 4, 6 there are entries, e.g. 80.00% instead of 80% (as in chart 5).

Response: The aesthetics of the charts has been improved in the manuscript.

4.The above comments do not interfere with the substantive content of the article. The scientific importance of the journal requires the authors to take into account the above comments.

Response: Thank you for your comments. The authors replied and added more content to the above comments.

5.The reviewer accepts the authors' explanations. The above comments are intended to improve the quality of the article. They are not obligatory.

Response: Thank you for your comments again. The authors replied and added more content to the above comments.